# The Role of Human Endogenous Retrovirus (HERV)-K119 *env* in THP-1 Monocytic Cell Differentiation

**DOI:** 10.3390/ijms242115566

**Published:** 2023-10-25

**Authors:** Eun-Ji Ko, Min-Hye Kim, Do-Ye Kim, Hyojin An, Sun-Hee Leem, Yung Hyun Choi, Heui-Soo Kim, Hee-Jae Cha

**Affiliations:** 1Departments of Parasitology and Genetics, Kosin University College of Medicine, Busan 49267, Republic of Korea; nebbia1127@gmail.com (E.-J.K.); kmhmary93@naver.com (M.-H.K.); dongkae97@gmail.com (D.-Y.K.); 322002@kosin.ac.kr (H.A.); 2Department of Biomedical Sciences, Dong-A University, Busan 49315, Republic of Korea; shleem@dau.ac.kr; 3Department of Biochemistry, College of Oriental Medicine, Dongeui University, Busan 47227, Republic of Korea; choiyh@deu.ac.kr; 4Department of Biological Sciences, College of Natural Sciences, Pusan National University, Busan 49241, Republic of Korea; khs307@pusan.ac.kr; 5Institute for Medical Science, Kosin University College of Medicine, Busan 49267, Republic of Korea

**Keywords:** HERV-K *env*, monocytic cell, CD32, immune response, CRISPR-Cas9

## Abstract

Human endogenous retrovirus (HERV)-K was reportedly inserted into the human genome millions of years ago and is closely related to various diseases, including cancer and immune regulation. In our previous studies, CRISPR-Cas9-enabled knockout (KO) of the HERV-K *env* gene was found to potentially reduce cell proliferation, cell migration, and invasion in colorectal and ovarian cancer cell lines. The immune response involves the migration and invasion of cells and is similar to cancer; however, in certain ways, it is completely unlike cancer. Therefore, we induced HERV-K119 *env* gene KO in THP-1, a monocytic cell that can be differentiated into a macrophage, to investigate the role of HERV-K119 *env* in immune regulation. Cell migration and invasion were noted to be significantly increased in HERV-K119 *env* KO THP-1 cells than in MOCK, and these results were contrary to those of cancer cells. To identify the underlying mechanism of HERV-K119 *env* KO in THP-1 cells, transcriptome analysis and cytokine array analysis were conducted. Semaphorin7A (SEMA7A), which induces the production of cytokines in macrophages and monocytic cells and plays an important role in immune effector cell activation during an inflammatory immune response, was significantly increased in HERV-K119 *env* KO THP-1 cells. We also found that HERV-K119 *env* KO THP-1 cells expressed various macrophage-specific surface markers, suggesting that KO of HERV-K119 *env* triggers the differentiation of THP-1 cells from monocytic cells into macrophages. In addition, analysis of the expression of M1 and M2 macrophage markers showed that M1 macrophage marker cluster of differentiation 32 (CD32) was significantly increased in HERV-K119 *env* KO cells. These results suggest that HERV-K119 *env* is implicated in the differentiation of monocytic cells into M1 macrophages and plays important roles in the immune response.

## 1. Introduction

Human endogenous retrovirus (HERV)-K was inserted into the human genome millions of years ago and has been implicated in various diseases, including cancer and immune regulation. Most HERVs are currently inactive and non-infectious owing to recombination, deletions, and mutations after insertion into the host genome [1,2]. HERV-K elements have been reportedly expressed and transcribed as proteins in certain diseases, including cancer. Numerous studies have associated HERV-K *env* with cancer. RNA levels of HERV-K *env* are significantly upregulated in breast cancer, prostate cancer, melanoma, and ovarian cancer [3,4,5,6,7,8]. Based on current reports, HERV-K *env* is a promising potential cancer biomarker and has been proposed as a target for immunotherapy [9]. HERV elements have not only been widely studied in the context of cancer, but they have also been reported to be related to various immune diseases. Particularly, HERVs are believed to play a pathogenic role in several autoimmune diseases, with a particular emphasis on rheumatic conditions such as rheumatoid arthritis (RA) and systemic lupus erythematosus (SLE) [10]. Various mechanisms have been described through which autoimmunity can be induced by HERVs. Among these, some exhibit greater biological plausibility, while others have more substantial supporting evidence. Multiple autoimmune diseases have been implicated in the pathogenesis involving HERVs [11]. Inflammatory diseases, often accompanied by autoimmune conditions, constitute a significant proportion of HERV-related disorders, apart from cancer. However, the mechanism through which HERV influences inflammatory diseases, including autoimmune disorders, remains unclear [12]. Another study reported that both the recombinant TM protein of HERV-K produced in yeast cells and a synthetic peptide corresponding to a conserved domain of the TM protein inhibit the proliferation of human and mouse immune cells. Additionally, they modulate the release of various cytokines, including the immunosuppressive IL-10, and alter the expression of numerous genes in human Peripheral Blood Mononuclear Cell (PBMCs). Furthermore, HERV-K particles produced by human teratocarcinoma cells were also found to induce the release of interleukin-10 (IL-10) [13]. HERV transactivation can contribute to inflammation, particularly due to the ability of the HERV-W family to interact with toll-like receptor (TLRs), as previously mentioned for TLR4 and CD14. This interaction leads to the induction of a robust pro-inflammatory response, including the release of several cytokines such as IL-1β, IL-6, and tumor necrosis factor-α (TNF-α) [14]. The HERV-K102 antigen and anti-HERV-K102 IgG from SLE patients form immune complexes that are readily phagocytosed by neutrophils, leading to the induction of NET formation. This process may contribute to autoimmunity and enhance interferon signaling in SLE [15]. Various factors expressed in the inflammatory response, either directly or indirectly, promote the expression of or activate the HERV gene, which has been suppressed through epigenetic regulation. In a previous study, HERV-K *env* RNA bound with MDA-5 and TLR3 receptors in THP-1 monocytic cells and induced the expression of type I interferons and cytokines, which confirmed that HERV-derived RNA participates in a selective signaling mechanism by binding to a receptor [16]. According to these hypotheses, activated HERV genes are likely to contribute to immune activity at both RNA and protein levels. The role of HERVs in cancer has been reported in previous studies. We found that HERV-K Env protein expression was significantly increased in 11 types of cancers, and knockout (KO) of the HERV-K *env* gene using CRISPR-Cas9 can potentially reduce tumorigenic characteristics, including proliferation, invasion, and migration, in DLD-1 colorectal cancer cells and ovarian cancer cell lines [17,18,19]. Another study demonstrated that the downregulation of HERV-K *env* RNA in pancreatic cancer cells using shRNA resulted in reduced cell proliferation and tumor growth via the RAS-ERK-RSK pathway [20]. However, the vital roles of HERV elements in immunity have not been confirmed through direct KO of the HERV-K119 *env* gene from immune cells. In this study, we examined the impact of HERV-K119 *env* KO in THP-1 and U937 monocytic leukemia cells to determine the role of HERV-K119 *env* in triggering an inflammatory response.

## 2. Results

### 2.1. Knockout of HERV-K119 env Gene in Human Monocytic Leukemia Cells

To identify the function of the HERV-K *env* gene in the immune response, HERV-K119 *env* KO THP-1 human monocytic leukemia cells were generated using the CRISPR-Cas9 gene editing system. Guide RNA was designed using the K119 region of the HERV-K *env* gene, which is the same guide RNA used in our previous study [18]. Genomic PCR was conducted using specific primers containing the guide RNA (gRNA)-selected region, which confirmed that the HERV-K *env* gene was removed (Figure 1A). RT-PCR results revealed that the HERV-K *env* gene was not expressed (Figure 1B). The RNA level of the HERV-K119 *env* gene was also confirmed using qRT-PCR and its expression significantly reduced at the RNA level in HERV-K119 *env* KO cells (Figure 1C). HERV-K env protein levels were significantly reduced in HERV-K env KO cells (Figure 1D). However, the expression of other HERV-K elements including *gag*, *pro*, and *pol* was not changed. The expression of *rec*, which is very close to that of *env*, was significantly reduced (Figure 1E). We also generated HERV-K119 *env* KO U937 human monocytic leukemia cells to support the results of HERV-K119 *env* KO in THP-1 cells and confirmed that the HERV-K119 *env* gene was knocked out in U937 cells using genomic PCR, RT-PCR, and Western blotting (Appendix A).

### 2.2. Effects of HERV-K119 env KO on Cytokine Secretion, Cell Proliferation, Migration, and Invasion in THP-1 Human Monocytic Leukemia Cells

To analyze the effect of HERV-K119 *env* KO on cytokine secretion, we conducted a cytokine array analysis in HERV-K119 *env* KO DLD-1 and HCT116 human colorectal cancer cells [18], and in HERV-K119 *env* KO THP-1 human monocytic leukemia cells. The effect of HERV-K119 *env* KO on the cytokine secretion patterns in THP-1 cells differed from that in colon cancer cell lines. Expressions of various cytokine factors increased in HERV-K119 *env* KO THP-1 cells, including CXCL-1, MIP-1α, MIP-1β, CCL5, and IL-8 (Figure 2A). The suspended THP-1 and U937 cells were treated with PMA to induce cell differentiation, followed by an LPS application to initiate inflammation. As shown in Figure 2, cell invasion and migration in HERV-K119 *env* KO cells significantly increased when MOCK cells were treated with PMA or PMA + LPS. Figure 2C shows a notable increase in HERV-K119 *env* KO cell growth. Additionally, HERV-K119 *env* KO cells eliciting inflammatory responses displayed a significant increase in cell growth. These findings led us to hypothesize that, unlike the patterns observed in previously documented cancer cells, HERV-K119 *env* KO within immune cells may act via pathways distinct from cancer-related traits. The expression levels of RNAs for inflammatory cytokine marker genes (TNF-Alpha, IL-33, IL-8) were analyzed using RT-PCR (Figure 2D). The expression of IL-33 significantly increased in HERV-K119 *env* KO cells when treated with PMA or PMA + LPS, as well as when left untreated. The expression of TNF-Alpha also showed a significant increase in HERV-K119 *env* KO cells when treated with PMA, whether or not they were treated with PMA + LPS. However, there was no significant change in TNF-Alpha expression between MOCK and HERV-K119 *env* KO cells when TNF-Alpha levels were maximized through treatment with PMA + LPS. IL-8 showed a slight increase in HERV-K119 *env* KO cells, but this increase was not statistically significant (Figure 2D).

### 2.3. Transcriptome Analysis of HERV-K119 env KO THP-1 Monocytic Leukemia Cells

To discern the functional mechanism of the HERV-K119 *env* gene within the immune system, we conducted a transcriptome analysis of HERV-K119 *env* KO THP-1 cells: untreated, PMA-treated, and PMA + LPS-treated groups. The transcription pattern of all groups was compared with their corresponding MOCK groups using RNA sequencing. Alterations in gene expression are depicted in the heatmap in Figure 3A. In pursuit of a pertinent target for the HERV-K env gene, we conducted analyses using Venn diagrams and categories within the HERV-K119 *env* KO and MOCK group cells (Figure 3B,C). As shown in Figure 3D, the expression fold changes (FCs) of the HERV-K119 *env* KO and MOCK group cells were determined using the ExDEGA program. Notably, among the genes correlated with the inflammatory response, we observed a substantial increase in the semaphorin 7a (*SEMA7A*) gene. Within this gene set, special attention was directed toward the *sema7a* gene, as it is one of the most prominent targets related to the function of the HERV-K *env* gene. This inference was drawn from the outcomes of NGS and bioinformatics analyses, coupled with the phenotypic alterations observed in HERV-K119 *env* KO. Notably, *SEMA7A* has been proposed as a potential immune regulator of other genes. Expression levels of the *SEMA7A* gene were elevated 1.151-fold in HERV-K119 *env* KO cells, 19.296-fold in PMA-treated HERV-K119 *env* KO cells, and 20.369-fold in PMA + LPS-treated HERV-K119 env KO cells (Table 1, Figure 3D).

### 2.4. Effects of SEMA7A on THP-1 Human Monocytic Leukemia Cell Lines

Because the *SEMA7A* gene is proposed to be a crucial target of the HERV-K *env* gene KO according to mRNA sequencing analysis, an overexpression vector for the *SEMA7A* gene was introduced into THP-1 monocyte leukemia cells to analyze its function in THP-1 cells. The *SEMA7A* expression was subsequently validated, and increased *sema7a* RNA levels were confirmed in *SEMA7A*-overexpressed THP-1 cells using qRT-PCR and RT-PCR (Figure 4A). The expression of *SEMA7A* at the RNA level decreased when cells were treated with PMA or PMA + LPS; however, it did not significantly change at the protein level (Figure 4B). However, the expression of sema7a at the RNA and protein levels significantly increased in HERV-K119 *env* KO THP-1 cells treated with PMA or PMA + LPS, and these data are consistent with the RNA-seq results (Figure 4B) and the densitometry results obtained from three independent experiments. The most prominent trait of HERV-K119 *env* gene KO cells, cell proliferation, was significantly enhanced with the overexpression of the *SEMA7A* gene (Figure 4C). *SEMA7A* overexpression significantly increased cell proliferation in both MOCK and HERV-K119 *env* KO THP-1 cells (Figure 4C). To confirm the function of *SEMA7A* in monocytic leukemia cells, we induced HERV-K119 *env* gene KO in another monocytic leukemia cell line, U937. Genomic KO of the HERV-K *env* gene and the expression of HERV-K *env* at the RNA and protein levels were confirmed (Appendix A). The expression of *SEMA7A* at RNA and protein levels was significantly increased in HERV-K119 *env* gene KO U937 cells (Appendix A), and cell proliferation significantly enhanced with the *SEMA7A* gene overexpression (Appendix A). These results suggest that the heightened expression of the *SEMA7A* gene due to HERV-K119 *env* KO promotes the growth of THP-1 and U937 cells. Furthermore, these results suggest an impact of HERV-K *env* on the inflammatory response.

### 2.5. Expression Level of Macrophage Markers in HERV-K119 env KO Monocytic Leukemia Cells

Among the various functions of SEMA7A, we focused on SEMA7A-mediated induction of M1/M2 macrophage differentiation. Thus, we hypothesized that in the absence of HERV-K *env* within monocytes, deactivated M1 or M2 macrophages are activated by *SEMA7A*, consequently triggering an inflammatory response (Figure 5A). To test this hypothesis, we analyzed M1 macrophage markers, including CD80, CD32, and CD64, and M2 macrophage markers, including CD68, CD206, and CD163, in HERV-K119 *env* KO THP-1 cells and compared them with those in HERV-K119 *env* KO OVCAR3 ovarian cancer cells. As shown in Figure 5A, CD32 was significantly upregulated in HERV-K119 *env* KO THP-1 cells, but not in HERV-K119 *env* KO OVCAR3 ovarian cancer cells (Figure 5A). Similarly, upon examining the *SEMA7A*-overexpressing cells, the expression of CD32 was found to be significantly increased (Figure 5B). Collectively, these results implied that the activation of M1 macrophages was induced by HERV-K119 *env* KO through *SEMA7A*. The KO of HERV-K119 env led to the elevated expression of *sema7a* within THP-1 and U937 cells. This signaling subsequently activated M1 macrophages and triggered cell proliferation, and it may be related to increased invasion, migration, and an augmented inflammatory response.

## 3. Discussion

Several HERV-derived elements play significant roles in immune diseases, including autoimmune disorders. Although the underlying mechanism of the HERV immunomodulatory action has not yet been elucidated, the promotion of diverse immune responses using new antigens through the activation of latent HERVs is possible, which elicits multiple immune responses. However, HERV-K can be represented by dozens or even hundreds of retroviral gene copies within the human genome, and these copies may contain elements that are not universally present in all individuals or populations [21]. We attempted to elucidate the biological functions by performing a KO of the human-specific HERV-K119 *env* sequence in its full-length form. Recombinant transmembrane (TM) proteins and peptides, corresponding to the immunosuppressive domain (ISD) of HERV-K, inhibit the proliferation of human immune cells and regulate the release of cytokines, similar to the ISD of HIV-1 [22]. HERV-K *env* has also been reported to interfere with immunity through interaction with CD4 [23]. Moreover, HIV-tat upregulates HERV-K expression, which interferes with the susceptibility to HIV [24]. HIV *tat* also upregulated the *HRES-1* endogenous retrovirus, which causes the downregulation of CD4 that hinders the reinfection of T cells by the exogenous retrovirus [25]. Nevertheless, in our experiments, we observed that CD4 expression in THP-1 cells remained largely unchanged despite the deletion of HERV-K119 *env*. A recent study showed that even in the case of T cells that are CD4-positive due to HIV infection, the expression pattern of HERV-K *env* appears to differ based on the site of HIV viral insertion [23]. Therefore, it is apparent that more research results are required to engage in a thorough discussion of this correlation. Although the results appear conflicting and appear to go through complex control mechanisms, these findings suggest that HERV-derived proteins play a significant role in the inflammatory response. Even though HERV-K and other HERV elements are assumed to play important roles in the immune response, few studies have been conducted to identify their functions by knocking out HERV elements from immune cells. Previously, we reported that HERV-K *env*-related NUPR1 may activate the NUPR1 pathway in DLD-1 colorectal cancer cells through CRISPR-Cas9-mediated KO of HERV-K *env* [18]. Furthermore, we knocked out HERV-K *env* to determine its function in ovarian cancer cells [19]. In this study, we performed HERV-K119 *env* KO in THP-1 and U937 human monocytic leukemia cells to analyze the role of HERV-K119 *env* in the immune response. In particular, recent papers lead to the conclusion that the THP-1 cell line possesses unique characteristics as a model for investigating and estimating the immune-modulating effects of compounds under both activated and resting conditions of the cells [26,27]. A distinctive feature of HERV-K *env* gene KO immune cells is that the opposite phenomenon occurs in HERV-K *env* KO cancer cells [18]. HERV-K119 *env* gene KO THP-1 cells secrete higher levels of cytokine factors than colon cancer cells. When the cells were treated with PMA and LPS to induce inflammatory reactions, HERV-K119 *env* KO accelerated cell proliferation, invasion, and migration. These results contrast with the observed trend that we have been experimenting with involving cancer cells with the knocked-out HERV-K119 *env*.

Transcriptome analysis was conducted to elucidate the counter-regulatory mechanisms and roles of HERV-K119 *env* in immune responses. Consequently, we identified *SEMA7A* as one of the most important target genes involved in the HERV-K *env*-mediated immune response [28]. *SEMA7A* is closely associated with the pathogenesis of various autoimmune diseases, inflammation-related diseases, and tumors. Additionally, *SEMA7A* regulates the proliferation, migration, invasion, lymphatic vessel formation, and angiogenesis of various types of cancer cells and controls the inflammatory functional phenotype of macrophages [28]. A recent study reported that *SEMA7A* controls the inflammatory phenotype of macrophages and regulates human macrophage chemotaxis and chemokinesis [29]. *SEMA7A* has also been reported to regulate cytokine-induced memory-like responses in human natural killer cells, and erythrocyte-derived *SEMA7A* has been shown to induce thrombotic inflammation in myocardial ischemia/reperfusion injuries [30,31]. Recent research has shown that macrophages in the basal state (M0) can transition into either the M1 or M2 activation state following its exposure to specific mediators [32], and *SEMA7A* controls M1 or M2 macrophage differentiation [29].

Recent studies have revealed that the expression of HERV-K in human pluripotent cells exhibits cell-form-related functions and adhesion, mediated through interactions with membrane-bound proteins. HERV-K is also implicated in the mTOR pathway, a signaling pathway associated with cell proliferation [33]. Also, the HERV-K Env protein may induce neurotoxicity by interacting with CD98HC and its associated secondary signaling pathways in patients with ALS [34]. Furthermore, our results suggest that there will be proliferation-like effects on the HERV-K119 *env* KO environment of THP-1 and U937 cells, and that increased cell proliferation induced by sema7a is also expected to be involved in the mTOR pathway.

Therefore, we examined various M1/M2 markers using qRT-PCR to determine whether HERV-K119 *env* KO is also linked to macrophage differentiation. We observed a significant increase in the M1 macrophage marker CD32 compared to THP-1 cells, but no change in HERV-K119 *env* KO ovarian cancer cell lines [19]. In this study, we discovered for the first time that HERV-K119 *env* KO affects monocytic leukemia cells; triggers cytokine secretion; and accelerates cell proliferation, invasion, and migration during inflammatory reactions. HERV-K119 *env* KO also induced *SEMA7A* gene expression to activate M1 macrophage differentiation (Figure 6). However, further studies are required to confirm this. First, the detailed mechanism by which HERV-K119 *env* KO promotes cell proliferation, invasion, and migration and increases *SEMA7A* gene expression in THP-1 cells to stimulate M1 macrophage differentiation remains unclear. To identify these mechanisms, the comparison of several other genes with HERV-K119 *env* KO in THP-1 cells and their analysis are necessary to discover genes involved in these phenomena. In addition, further studies are required to determine the function of HERV-K119 *env* KO-induced *SEMA7A* in monocytic leukemia cells. It is also necessary to clarify whether *SEMA7A* is involved in cell division, invasion, and migration, phenomena other than M1 macrophage differentiation, and whether only *SEMA7A* or other factors are involved.

## 4. Materials and Methods

### 4.1. Cell Culture and Transfection

Human colorectal cancer cell lines DLD-1 and HCT116 were obtained from the American Type Culture Collection (ATCC, Manassas, VA, USA). DLD-1 cells were cultured in Dulbecco’s modified Eagle’s medium (DMEM) containing 10% fetal bovine serum (FBS; Invitrogen, Carlsbad, CA, USA), 1% penicillin, and L-glutamine (Thermo Fisher Scientific, Rockford, IL, USA). THP-1 and U937 cells were obtained from the Korean Cell Line Bank (Seoul, Republic of Korea). HCT-1, THP-1, and U937 cells were cultured in RPMI 1640 medium (Thermo Fisher Scientific) containing 10% FBS (Invitrogen), 1% penicillin, and L-glutamine (Thermo Fisher Scientific). All cell lines were maintained at 37 °C in a humidified atmosphere containing 5% CO_2_ and 90% humidity. Transfection of plasmids or gRNA was conducted using the Lipofectamine 2000 reagent (Thermo Fisher Scientific) according to the manufacturer’s instructions. Briefly, cells were trypsinized, counted, and seeded onto plates the day before transfection to ensure approximately 80% cell confluence on the day of transfection. The transfection efficiency was monitored using RT-PCR.

### 4.2. Generation of Knockout Cell Line with CRISPR-Cas9

Guide RNA sequences for CRISPR-Cas9 were designed using the CRISPR design provided by Toolgen, a company dedicated to CRISPR-Cas9 technology (Seoul, Republic of Korea). Insert oligonucleotide for HERV-K119 Env gRNA #3 was 5′-TATTCCAGTCACACTGTAACTGG-3′. The HERV-K *env* guide RNA targeted chr12:58,721,197-58,722,612 (−) of HERV-K119 *env* gene. The complementary oligonucleotides for gRNAs were annealed and cloned into the CRISPR/Cas9-Puro and pRGEN-Cas9-CMV/T7-Hygro-EGFP vectors (Toolgen) [18]. The THP-1 and U937 cells were transfected with CRISPR/Cas9+gRNA #3. A period of 18 hours after electroporation, the cells were treated with 100 μg/mL of hygromycin for 2 d. After two weeks, colonies were isolated using cloning cylinders for further investigation, including functional studies.

### 4.3. Plasmid Construct for Over-Expression

We purchased the *semea7a* over-expression vector. *Sema7a* gene was inserted into pcDNA3.1+/C-(K)-DYK expression vector and full sequences were confirmed through sequencing. The plasmid construct used in the present study was SEMA7A (Cat. NO OHu23073D) and was purchased from Genscript (Piscataway, NJ, USA). Overexpression plasmid transfection was performed using the Lipofectamine 2000 reagent (Thermo Fisher Scientific) and according to the Lipofectamine 2000 protocol.

### 4.4. RT-PCR, qRT-PCR and Genomic PCR (Polymerase Chain Reaction)

Total RNA was isolated from the cells using TRIzol reagent (Invitrogen), according to the manufacturer’s instructions. cDNA was synthesized using a PCR Mix (Bioneer, Seoul, Republic of Korea) to measure HERV-K gene expression. The primer sequences for the HERV-K element genes were as follows: HERV-K *env* sense: 5′-CAC AAC TAA AGA AGC TGA CG-3′ and HERV-K *env* antisense. GAPDH was used as a control (sense primer, 5′-CAA TGA CCC CTT CAT TGA CC-3′; antisense primer, 5′-GAC AAG CTT CCC GTT CTC AG-3′). HERV-K universal sense, 5’-GTG ACT GGA ATA CGT CAG ATT TTT G-3’; HERV-K119 antisense, 5’-TGA CCC CTG TCA CTC TAG TAA-3’; HERV-K *gag* sense, 5’-GAG AGC CTC CCA CAG TTG AG-3’; HERV-K *gag* antisense, 5’-TTT GCC AGA ATC TCC CAA TC-3’; HERV-K *pro* sense, 5’-TGG CCT AAA CAA AAG GCT GT-3’; HERV-K *pro* antisense, 5’-CGA CCC CAC AGA TTA AGA GG-3’; HERV-K *pol* sense, 5’-TTG AGC CTT CGT TCT CAC CT-3’; HERV-K *pol* antisense, 5’-CTG CCA GAG GGA TGG TAA AA-3’; HERV-K *rec* sense, 5’-ATC GAG CAC CGT TGA CTC ACA AGA-3’; HERV-K *rec* antisense, 5’-GGT ACA CCT GCA GAC ACC ATT GAT-3’ [35]. IL-33 sense, 5’-GTG ACG GTG TTG ATG GTA AGA T; IL-33 antisense, 5’-AGC TCC ACA GAG TGT TCC TTG-3’; IL-8 sense, 5’-ATG ACT TCC AAG CTG GCC GTG GCT; IL-8 antisense, 5’-TCT CAG CCC TCT TCA AAA ACT TCT C-3’. RT-PCR cycling conditions were 94 °C for 2 min to activate DNA polymerase; followed by 35 cycles of 94 °C for 1 min, 58 °C for 1 min, and 72 °C for 1 min; and 72 °C for 10 min for post-elongation. TNF-Alpha sense, 5’- CCA CAC CAT CAG CCG CAT CG-3’; TNF-Alpha anti sense, 5’-GGG CCG ATT GAT CTC AGC GC-3’. RT-PCR cycling conditions were 94 °C for 2 min to activate DNA polymerase; followed by 33 cycles of 94 °C for 30 s, 64 °C for 30 s, and 72 °C for 30 s; and 72 °C for 10 min for post-elongation. The products were electrophoresed on a 1.8% agarose gel and photographed under LED light. qRT-PCR was performed using TB Green Premix Taq (Takara, Japan) and the results were recorded using QuantStudio 3 (Thermo Fisher Scientific). The relative gene expression levels were quantified based on the 2^−ΔΔCt^ method and normalized to the reference gene. The primer sequences used for qRT-PCR are listed in Table 2.

### 4.5. Western Blot Analysis

Western blot analysis was performed as previously described [41]. Briefly, 50 μg of protein extract was prepared using PRO-PREP Protein Extraction Solution (Intron Biotechnology, Kyunggi, Republic of Korea) and separated via electrophoresis on a Novex 4–12% Bis-Tris gel (Invitrogen). Protein concentrations were determined using a bicinchoninic acid protein assay system (Pierce, Rockford, IL, USA) and equal amounts of each sample were separated via electrophoresis on Novex 4–12% Bis-Tris gels. Equal protein loading was confirmed through Coomassie Blue staining of duplicate gels after electrophoresis. The gels were incubated in blotting buffer containing 1× NuPAGE® Bis-Tris transfer buffer (Invitrogen) and 20% methanol for 30 min at room temperature. Proteins were transferred to nitrocellulose membranes (Invitrogen) via electrotransfer. Membranes were preincubated for 2 h in Tris-buffered saline (TBS) containing 5% skim milk and 0.05% Tween 20 (TBS-T). The membranes were incubated overnight at 4 °C in TBS-T with each antibody. The antibodies used and their dilution were as follows: HERV-K Env (1:2000; Austral Biologicals, San Ramon, CA, USA) and glucose 6 phosphate dehydrogenase (GAPDH, 1:5000; R&D Systems, Minneapolis, MN, USA). The membranes were washed five times with TBS-T and immunoreactive proteins were detected using ECL Western blotting detection reagents (Amersham, RPN2106, Buckinghamshire, UK).

### 4.6. Cytokine Assay

A sample (DLD-1, DLD-1 HERV-K env KO, HCT116, HCT116 HERV-K env KO, THP-1, and THP-1 HERV-K env ko cells) was prepared 24 h prior and cultured in conditional media. The Proteome Profiler Array (Human Cytokine Array, abcam, ARY005B, Cambridge, UK) kit was used. The experiment was conducted following the manufacturer’s protocol.

### 4.7. Invasion and Migration Assays

In vitro migration and invasion assays were performed as previously described [42]. Briefly, Transwell chambers containing membranes with an 8 μm pore size (Invitrogen) were used for both assays. For the migration assay, 600 μL of conditioned medium obtained by culturing THP-1 cells (PMA, PMA + LPS, KO, KO PMA, and KO PMA + LPS) for 18 h in serum-free RPMI was placed into the lower chambers of each well. THP-1 cells (1 × 10^4^ cells) were resuspended in 100 μL of serum-free RPMI and placed in the upper chambers of each well. The chambers were incubated for 18 h at 37 °C, and the cells in the lower chambers were fixed and stained with Diff Quit (Sysmex, Japan), according to the Diff Quit protocol. The invasion assay was performed in a similar fashion, except that the upper surface of the Transwell filter was coated with 20 µL of 0.5 mg/mL Matrigel (BD Biosciences, Bedford, MA, USA) before the cells were added to the upper chambers. All experiments were repeated at least thrice, and each data point was measured in triplicate. Mean values and 95% confidence intervals (CIs) were calculated.

### 4.8. Cell Viability Assay

For the cell viability assay, cells (1 × 10^4^ cells per well) were plated in complete medium in a 6-well plate and incubated for 72 h. Cells were then harvested, and viable cells were counted using the trypan blue dye exclusion method to determine the cell proliferation rates.

### 4.9. RNA-Seq Data Analysis

Libraries were prepared from total RNA using the NEBNext Ultra II Directional RNA-Seq Kit (NEW ENGLAND BioLabs Inc., Hitchin, UK). mRNA was isolated using a poly (A) RNA Selection Kit (LEXOGEN Inc., Vienna, Austria). The isolated mRNAs were used for cDNA synthesis and shearing, according to the manufacturer’s instructions. Indexing was performed using Illumina indexes 1–12. Enrichment was performed using PCR. Subsequently, libraries were examined using a TapeStation HS D1000 Screen Tape (Agilent Technologies, Amstelveen, The Netherlands) to evaluate the mean fragment size. Quantification was performed using a library quantification kit and StepOne Real-Time PCR System (Life Technologies Inc., Carlsbad, CA, USA). High-throughput sequencing was performed via paired-end 100 sequencing using NovaSeq 6000 (Illumina, Inc., San Diego, CA, USA). The transcriptome analysis was performed using Ebiogen (Ebiogen Inc., Seoul, Republic of Korea). Quality control of the raw sequencing data was performed using Fast QC. Adapter and low-quality reads (<Q20) were removed using FASTX_Trimmer and BBMap version 38.79. Trimmed reads were mapped to the reference genome using TopHat software version 2.1.0. Read Count (RC) data were processed based on the Fragments per kilobase per million (FPKM + G) + geometric normalization method using EdgeR within R (version 3.20.1). Fragments per kilobase per million (FPKM) reads were estimated using Cufflinks. Data mining and graphic visualization were performed using ExDEGA (Ebiogen Inc., Seoul, Republic of Korea) [43,44].

### 4.10. Statistical Analysis

Descriptive statistics were calculated for the patients’ characteristics. Statistical significance was defined as a two-tailed *p*-value < 0.05. The fluorescence intensity was read using imageJ analysis software (version 1.53) and the intensity was measured to calculate the mean values and 95% CIs. The statistical significance of differences among groups was determined using a two-tailed Student’s *t*-test.

## 5. Conclusions

In conclusion, our results highlighted the significance of HERV-K119 *env* KO in monocytic leukemia cells. In accordance with the pathway delineated in Figure 6, HERV-K119 *env* KO of this gene affected monocytic leukemia cells; triggered cytokine secretion; and accelerated cell proliferation, invasion, and migration upon the induction of inflammatory reactions. Additionally, the knockout of this gene induced the expression of the *sema7a* gene, leading to the activation of M1 macrophage differentiation.

## Figures and Tables

**Figure 1 ijms-24-15566-f001:**
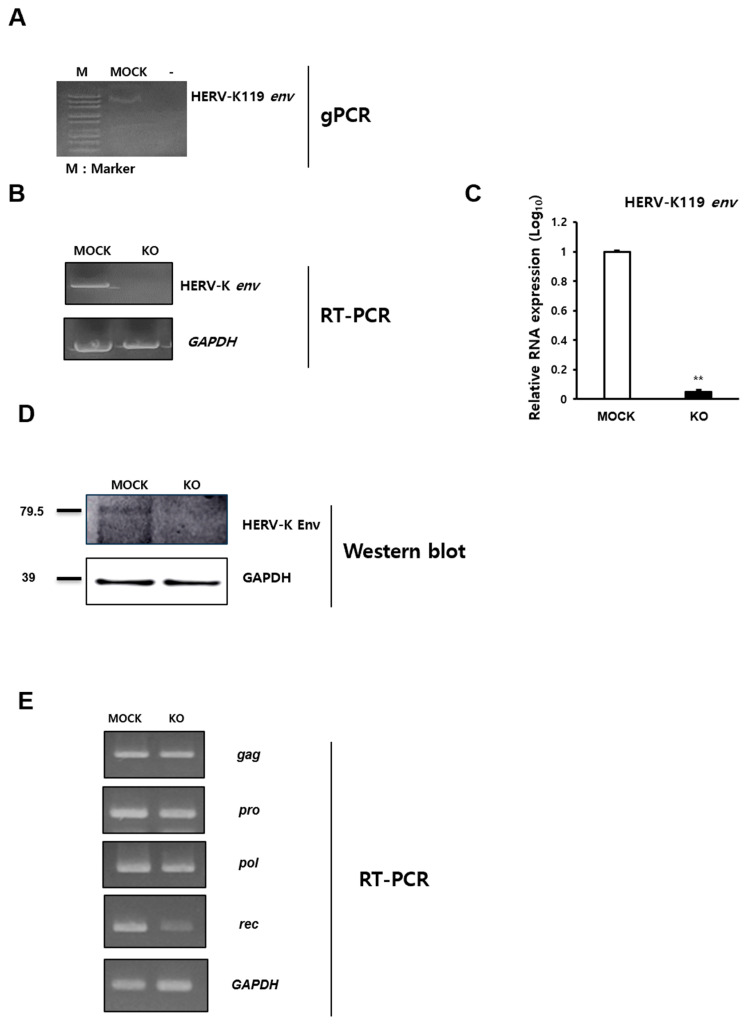
Knockout of HERV-K119 *env* gene and expression levels of its RNA and protein in HERV-K119 *env* KO THP-1 monocytic leukemia cells. (**A**) Knockout of HERV-K env gene in HERV-K119 region. Genomic polymerase chain reaction (gPCR) was performed for specific regions of HERV-K119 env gene to confirm its KO in THP-1 monocytic leukemia cells. (**B**) Expression of HERV-K *env* gene in HERV-K119 *env* KO THP-1 cells. The expression of HERV-K *env* at RNA level was analyzed using RT-PCR. RT-PCR performed for the general region of HERV-K *env* gene. (**C**) HERV-K119 *env* at RNA level was analyzed using real-time qPCR. (**D**) Expression of HERV-K Env protein in HERV-K119 *env* KO THP-1 cells. Western blotting was performed to analyze the protein level of HERV-K Env. (**E**) Expression of HERV-K119 *gag*, *pro*, *pol* and *rec* genes in THP-1 monocytic leukemia cells. **, *p* < 0.01; *n* ≥ 3 per group.

**Figure 2 ijms-24-15566-f002:**
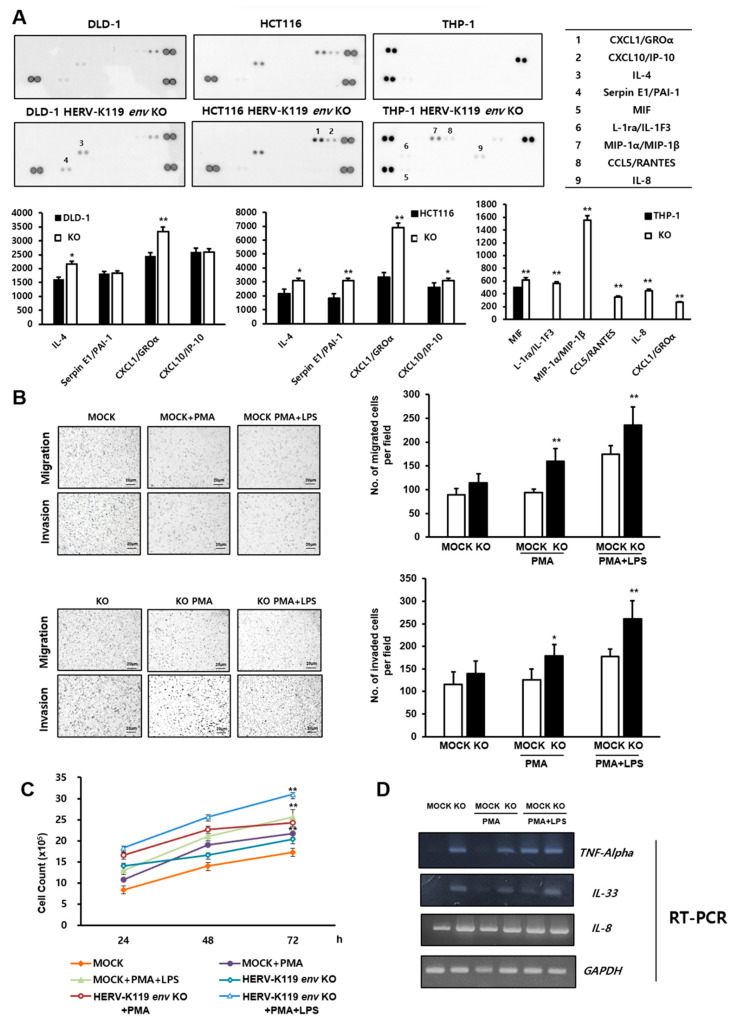
Effects of HERV-K119 *env* KO on cytokine secretion, proliferation, migration, and invasion in THP-1 human monocytic leukemia cells. (**A**) The human cytokine array analysis of HERV-K119 *env* KO cells for the detection of cytokines, chemokines, and growth factors. The expression of diverse cytokine factors in HERV-K119 *env* KO cells was verified. (**B**) Invasion and migration of HERV-K119 *env* KO THP-1 cells. The invasion and migration were significantly increased in PMA- or PMA + LPS-treated HERV-K119 *env* KO cells. (**C**) Cell proliferation assay of HERV-K119 *env* KO THP-1 cells. The effect of HERV-K119 *env* KO on cell proliferation was observed in the HERV-K119 *env* KO THP-1 cells as well as in the PMA and PMA + LPS treatment groups. A notable increase was observed in cell proliferation within the HERV-K119 *env* KO group and a particularly heightened increase was observed in the groups induced with inflammation. (**D**) RNA levels of inflammatory cytokines including TNF-alpha, IL-33, and IL-8 through RT-PCR analysis. KO, HERV-K119 *env* KO. *, *p* < 0.05, **, *p* < 0.01; *n* ≥ 3 per group.

**Figure 3 ijms-24-15566-f003:**
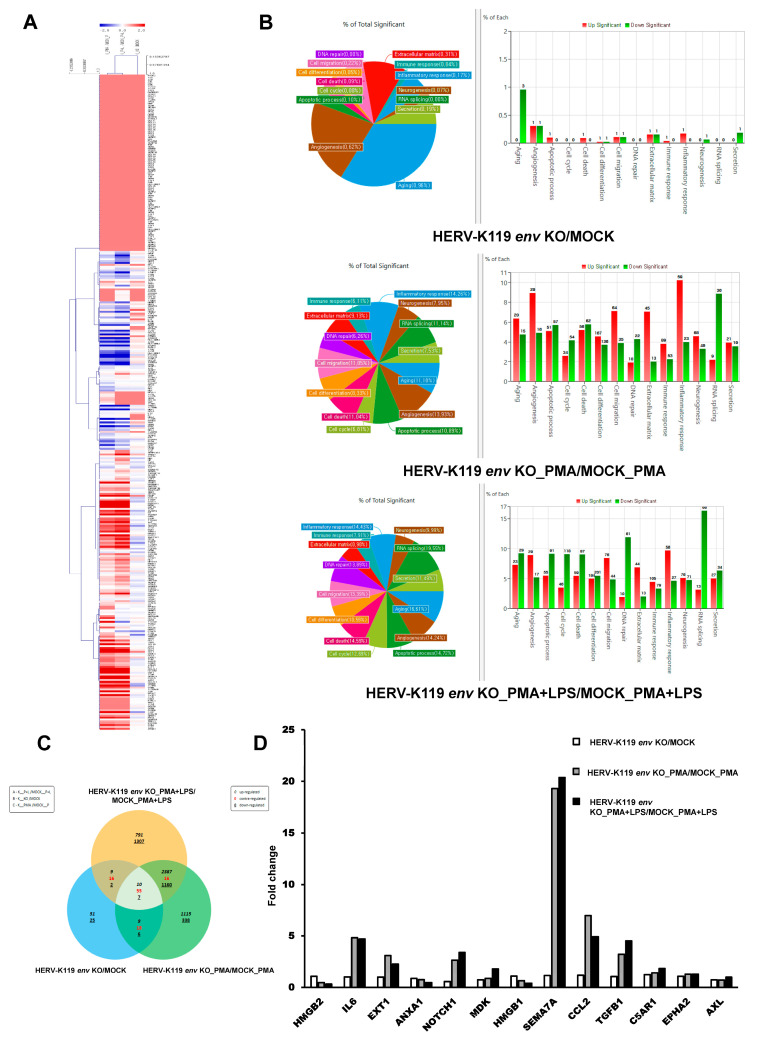
Transcriptome analysis of HERV-K119 *env* KO THP-1 and MOCK group cells. (**A**) Heatmap of changes in gene expressions in THP-1 HERV-K119 *env* KO and MOCK groups cells. (**B**) Gene ontology analysis of differentially expressed genes (DEGs) in HERV-K119 *env* KO and MOCK cells. (**C**) Venn diagrams and (**D**) Fold changes of THP-1 HERV-K119 *env* KO and MOCK groups cells.

**Figure 4 ijms-24-15566-f004:**
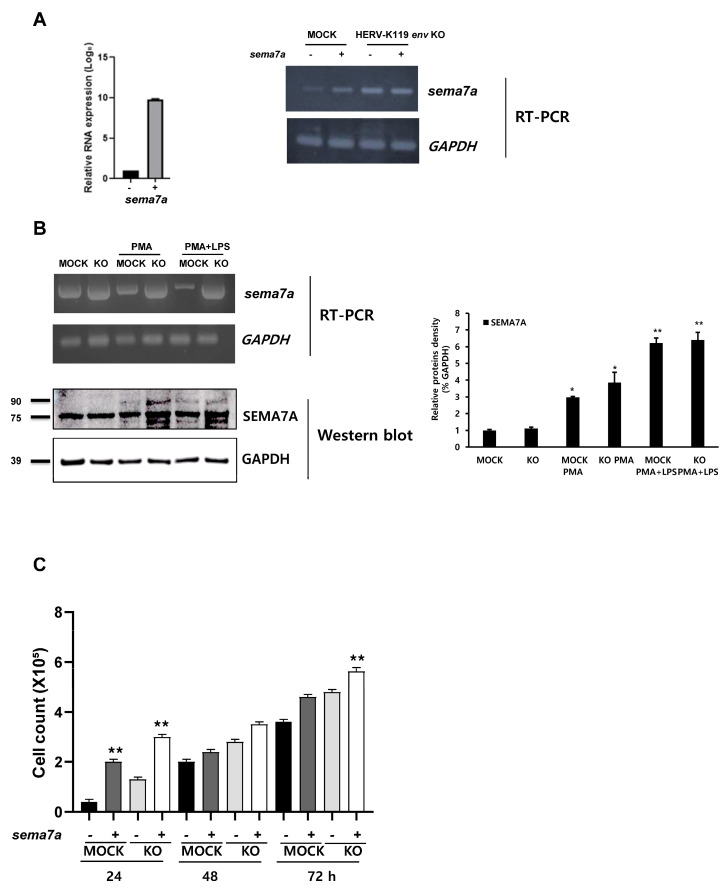
Effect of *SEMA7A* gene on HERV-K119 *env* KO THP-1 monocyte leukemia cells. (**A**) RNA expression of *SEMA7A* gene in *SEMA7A*-overexpressing THP-1 cells. Real-time qPCR (left) and RT-PCR (right) performed to confirm the expression of *sema7a* gene in *sema7a*-overexpressing THP-1 cells. (**B**) Expression of *SEMA7A* gene at the RNA and protein levels. RNA and protein expression of *SEMA7A* in MOCK and HERV-K119 *env* KO THP-1 cells: untreated and treated with PMA or PMA + LPS. (**C**) Effect of *SEMA7A* gene overexpression on THP-1 cell proliferation. The cell proliferation rate was significantly increased through overexpression of the *SEMA7A* gene in THP-1 cells. KO, HERV-K119 *env* KO; *, *p* < 0.05, **, *p* < 0.01; *n* ≥ 3 per group.

**Figure 5 ijms-24-15566-f005:**
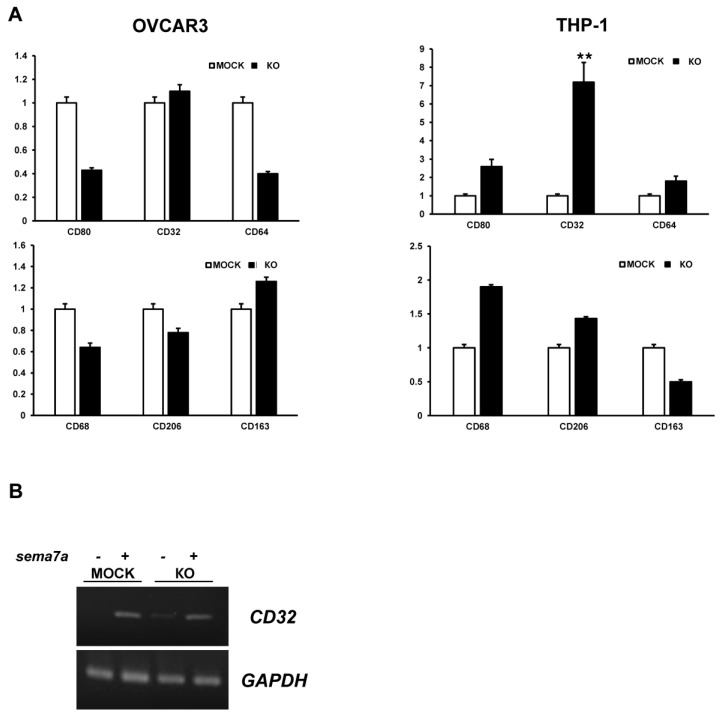
Effect of HERV-K119 *env* KO on M1/M2 macrophage differentiation. (**A**) Expression of M1/M2 macrophage markers in HERV-K119 *env* KO OVCAR3 ovarian cancer cells and HERV-K119 *env* KO THP-1 cells. (**B**) Expression of M1 macrophage marker CD32 in *SEMA7A*-overexpressing THP-1 cells. -, empty pcDNA vector; +, SEMA7A overexpression; KO, HERV-K119 *env* KO. **, *p* < 0.01; *n* ≥ 3 per group.

**Figure 6 ijms-24-15566-f006:**
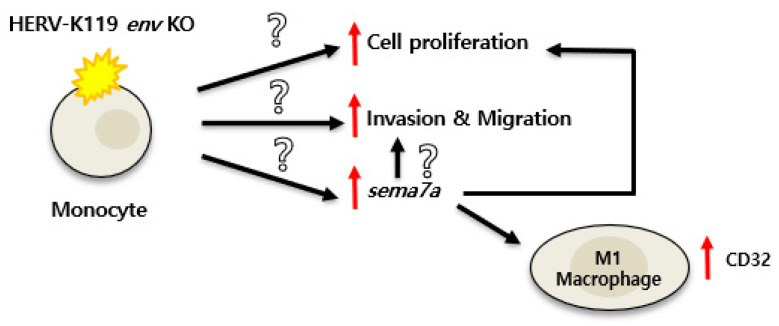
Schematic diagram of the immune response related to the HERV-K119 *env* KO. HERV-K119 *env* KO affected monocytic leukemia cells; triggered cytokine secretion; and accelerated cell proliferation, invasion, and migration upon inflammatory reaction induction. HERV-K119 *env* KO also induced *SEMA7A* expression to activate M1 macrophage differentiation. ↑: upregulation/?: Currently unknown.

**Table 1 ijms-24-15566-t001:** Differential gene expression in SEMA7A by HERV-K119 *env* KO and treated cells.

Target Gene	HERV-K *env* KO	Fold Change (/MOCK, Treatment Groups)	Classification	Raw Data (FPKM)
*SEMAA7A*	K *env* KO Treated PMA Treated PMA + LPS	1.15 19.296 20.369	Up-regulation	145 1185 1299

**Table 2 ijms-24-15566-t002:** Primer sequences used for real-time qRT-PCR.

Gene	Sense	Antisense	Ref.
*Sema7a*	ACAGGGGCACTATCCACAAG	CTCAGCATCCAGCGACAT	
*CD80*	CACCTGGCTGAAGTGAC	GTCAGGCAGCATATCAC	Duan et al. [36]
*CD86*	GGGCCGCACAAGTTTTGA	GCCCTTGTCCTTGATCTGAA	Duan et al. [36]
*CD64*	ATGGCACCTACCATTGCTCAGG	CCA AGCACTTGAAGCTCCAACTC	Zhang et al. [37]
*CD32*	AATCCTGCCGTTCCTACTGATC	GTGTCACCGTGTCTTCCTTGAG	Ye et al. [38]
*CD206*	GTTCACCTGGATGATGGTTCTC	AGGACATGCCAGGGTCACCTTT	Manuelpillai et al. [39]
*CD163*	CAGGAAACCAGTCCCAAACA	AGCGACCTCCTCCATTTACC	Alshahrani et al. [40]
*CD68*	GCTACATGGCGGTGGAGTACAA	ATGATGAGAGGCAGCAAGATGG	Alshahrani et al. [40]
*GAPDH*	TGTTCCTACCCCCAATGTGT	TGTGAGGGAGATGCTCAGTG	Manuelpillai et al. [39]

## Data Availability

The data generated in this study are publicly available in Gene Expression Omnibus (GEO) at GSE240644.

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
