# Peer review of "The Role of Human Endogenous Retrovirus (HERV)-K119 env in THP-1 Monocytic Cell Differentiation"

_ijms, 2023, doi:10.3390/ijms242115566_

Round 1

Reviewer 1 Report

The manuscript from Eun-Ji Ko et al. presents interesting and meaningful results from the study of the role of HERV-K119 env gene in monocyte differentiation, using THP-1 cell line as a model.

The conclusion of a role of HERV-K(119!) env is however made from a “negative” point of view: K(119) env Ko THP-1 cells display pro-inflammatory cytokine production and M1 macrophage differentiation. The “positive” counterpart is missing since, as already published in other cells and conditions (Wang et al. 2020. Regulation of stem cell function and neuronal differentiation by HERV-K via mTOR pathway. Proc Natl Acad Sci U S A 117:17842-17853), this should implicitly show that HERV-K env is involved in maintaining the undifferentiated and proliferating state of monocytic THP-1 (and also U937) cells.

This study therefore provides important results confirming and widening the role of HERV-K in counteracting cell differentiation and in promoting their proliferation in this undifferentiated state. This was shown to be mediated by an interaction with CD98HC, which may reveal harmful when this interaction occurs in differentiated and functional cells (Steiner et al. 2022. Human Endogenous Retrovirus K Envelope in Spinal Fluid of Amyotrophic Lateral Sclerosis Is Toxic. Ann Neurol 92:545-561).

However, the general understanding of HERV biology and genetics as reflected by the references, the introduction and the interpretation, is missing or does not reflect major features of this scientific domain:

-HERVs, and HERV-K in particular, represent families of multicopy proviral elements in the human genome. This may represent tens or hundreds of retroviral gene copies in the human genome and may also comprise elements that are not present in every individual or population (Wildschutteet al. 2016. Discovery of unfixed endogenous retrovirus insertions in diverse human populations. Proc Natl Acad Sci U S A 113:E2326-2334). Therefore, because the “Guide RNA was designed using the K119 region of the HERV-K env gene”, the authors can only claim to have HERV-K119 KO cells, and not “HERV-K env” cells, for they just cannot have eliminated all other env copies from the many other KERV-K inserts in human cells.

It should also be specified that the validation of K119-env was confirmed at the transcriptional level and not at the genomic level (DNA), for this would have shown many other remaining K-env copies. This is understandable for practical reasons but must be explained.

Specific remarks:

Title: should specify HERV-K119

Introduction:

-many references are old, not always accurate with up-to-date knowledge, e.g., ref.10, 13,14... Should be “updated”.

-the assertion “Recent reports have identified the HERV locus as a potential regulator of autoimmunity” is not correct. It would mean one HERV locus among hundreds (rather thousands taken altogether), which is not likely to be reliable and should anyhow specify which locus while presenting it as one hypothetical HERV copy involved in autoimmunity. It just cannot be presented as presently written, citing publications of low relevance and rather outdated.

- concluding “we examined the impact of HERV-K env KO in THP-1  and U937 monocytic leukemia cells to determine the correlation between HERV-K  env and the immune response, focusing on its role in triggering an inflammatory response” is not correct for the immune response, as said here, has not be examined. The sentence would be correct with the “focus”, such as: “to determine HERV-K  env role in triggering an inflammatory response”

-Results:

As said, it should be mentioned that the K119-env KO status is checked from the transcriptional level (specific RNA expression), maybe also needing to explain how the RT-PCR is made selective (or whether only K119-env is transcribed in these cells ?).

Figure 1:

Photo legends in A and B and C should be the same: “HERV-K119 env”.

In C, international standards now require to show the photo from the complete WB gels.

Figure 2.

I could not see anything clear in photos. The quality is too low and most readers downloading PDF version will not see what is supposed to be shown either.

In c: differences in cell counts are modest, which makes the number of replicates important to specify to understand whether this is really significant or only a weak trend. (sextuplicates would be needed here)

-when written “Because the sema7a gene is proposed to be a crucial target of the HERV-K env gene” may you cite the publications that reliably and consistently supported this assertion? Otherwise, it may be better not to say it so.

Figure 3:

The quality in PDF is too low to understand anything (this points to the quality of the original picture). Only SEMA7A increased transcription ), along with IL-6, CCL2 and TGFb1 (not “expression”, which is scientifically too vague or would suggest that this is a protein study), are visible.

Diagrams should also be better described and explained in the legend.

Discussion:

-Issues in the following sentences:

“The surface (SU) region of the HERV Env protein induces cell–cell fusion and the formation of fusions, leading to tumorigenesis and chromosomal instability” and “The TM region of HERV Env facilitates immune evasion by tumor cells through immunosuppressive activity”.

This is not true as said, since many differences exist between envelope proteins from the different HERV families and also between the different copies from the same family. These are false assertions as written.

“ These findings suggest that HERV-derived proteins play a significant role in inflammatory response.”

This study on HERV-K119 env shows the reverse! It shows that HERV-K119 env KO monocytic cells display a strong inflammatory response, which implicitly means that HERV-K119 env should prevent this pro-inflammatory phenotype, probably by preventing their differentiation into macrophages and maintaining them in a proliferative undifferentiated state. This is a misunderstanding or a very confusing copy-paste mistake. Please provide the correct scientific conclusion.

minor corrections only.

Author Response

The manuscript from Eun-Ji Ko et al. presents interesting and meaningful results from the study of the role of HERV-K119 env gene in monocyte differentiation, using THP-1 cell line as a model.

The conclusion of a role of HERV-K(119!) env is however made from a “negative” point of view: K(119) env Ko THP-1 cells display pro-inflammatory cytokine production and M1 macrophage differentiation. The “positive” counterpart is missing since, as already published in other cells and conditions (Wang et al. 2020. Regulation of stem cell function and neuronal differentiation by HERV-K via mTOR pathway. Proc Natl Acad Sci U S A 117:17842-17853), this should implicitly show that HERV-K env is involved in maintaining the undifferentiated and proliferating state of monocytic THP-1 (and also U937) cells.

This study therefore provides important results confirming and widening the role of HERV-K in counteracting cell differentiation and in promoting their proliferation in this undifferentiated state. This was shown to be mediated by an interaction with CD98HC, which may reveal harmful when this interaction occurs in differentiated and functional cells (Steiner et al. 2022. Human Endogenous Retrovirus K Envelope in Spinal Fluid of Amyotrophic Lateral Sclerosis Is Toxic. Ann Neurol 92:545-561).

However, the general understanding of HERV biology and genetics as reflected by the references, the introduction and the interpretation, is missing or does not reflect major features of this scientific domain:

-HERVs, and HERV-K in particular, represent families of multicopy proviral elements in the human genome. This may represent tens or hundreds of retroviral gene copies in the human genome and may also comprise elements that are not present in every individual or population (Wildschutteet al. 2016. Discovery of unfixed endogenous retrovirus insertions in diverse human populations. Proc Natl Acad Sci U S A 113:E2326-2334).

Therefore, because the “Guide RNA was designed using the K119 region of the HERV-K env gene”, the authors can only claim to have HERV-K119 KO cells, and not “HERV-K env” cells, for they just cannot have eliminated all other env copies from the many other KERV-K inserts in human cells.

: We corrected and changed HERV-K to HERV-K119.

It should also be specified that the validation of K119-env was confirmed at the transcriptional level and not at the genomic level (DNA), for this would have shown many other remaining K-env copies. This is understandable for practical reasons but must be explained.

: Our result of qRT-PCR (Fig. 1C) is performed with specific primer for HERV-K119 and RT-PCR (Fig. 1B) is for general env gene. We clarified it and correct contents and figure.

Specific remarks:

Title: should specify HERV-K119

Introduction:

-many references are old, not always accurate with up-to-date knowledge, e.g., ref.10, 13,14... Should be “updated”.

: We corrected and changed HERV-K to HERV-K119. We updated references.

-the assertion “Recent reports have identified the HERV locus as a potential regulator of autoimmunity” is not correct. It would mean one HERV locus among hundreds (rather thousands taken altogether), which is not likely to be reliable and should anyhow specify which locus while presenting it as one hypothetical HERV copy involved in autoimmunity. It just cannot be presented as presently written, citing publications of low relevance and rather outdated.

: We corrected contents and removed sentence.

- concluding “we examined the impact of HERV-K env KO in THP-1  and U937 monocytic leukemia cells to determine the correlation between HERV-K  env and the immune response, focusing on its role in triggering an inflammatory response” is not correct for the immune response, as said here, has not be examined. The sentence would be correct with the “focus”, such as: “to determine HERV-K  env role in triggering an inflammatory response”

: We corrected

-Results:

As said, it should be mentioned that the K119-env KO status is checked from the transcriptional level (specific RNA expression), maybe also needing to explain how the RT-PCR is made selective (or whether only K119-env is transcribed in these cells ?).

: As we answered previously, our result of qRT-PCR is performed with specific primer for HERV-K119 and RT PCR is for general env gene. We added primer sequences for RT-PCR and qRT-PCR.

Figure 1:

Photo legends in A and B and C should be the same: “HERV-K119 env”.

In C, international standards now require to show the photo from the complete WB gels.

: We corrected and added.

Figure 2.

I could not see anything clear in photos. The quality is too low and most readers downloading PDF version will not see what is supposed to be shown either.

In c: differences in cell counts are modest, which makes the number of replicates important to specify to understand whether this is really significant or only a weak trend. (sextuplicates would be needed here) : We corrected and added.

-when written “Because the sema7a gene is proposed to be a crucial target of the HERV-K env gene” may you cite the publications that reliably and consistently supported this assertion? Otherwise, it may be better not to say it so. :

: We corrected sentence described below.

Because the sema7a gene is proposed to be a crucial target of the HERV-K env gene KO according to analysis of mRNA sequencing

Figure 3:

The quality in PDF is too low to understand anything (this points to the quality of the original picture). Only SEMA7A increased transcription ), along with IL-6, CCL2 and TGFb1 (not “expression”, which is scientifically too vague or would suggest that this is a protein study), are visible.

Diagrams should also be better described and explained in the legend. :

: We corrected and added.

Discussion:

-Issues in the following sentences:

“The surface (SU) region of the HERV Env protein induces cell–cell fusion and the formation of fusions, leading to tumorigenesis and chromosomal instability” and “The TM region of HERV Env facilitates immune evasion by tumor cells through immunosuppressive activity”.

This is not true as said, since many differences exist between envelope proteins from the different HERV families and also between the different copies from the same family. These are false assertions as written.

: We deleted these contents.

“ These findings suggest that HERV-derived proteins play a significant role in inflammatory response.”

This study on HERV-K119 env shows the reverse! It shows that HERV-K119 env KO monocytic cells display a strong inflammatory response, which implicitly means that HERV-K119 env should prevent this pro-inflammatory phenotype, probably by preventing their differentiation into macrophages and maintaining them in a proliferative undifferentiated state. This is a misunderstanding or a very confusing copy-paste mistake. Please provide the correct scientific conclusion.

: We agree that certain HERV elements are elevated in response to inflammation. However, we cannot definitively conclude that these results imply that HERV elements themselves are responsible for triggering the inflammatory response. Our findings demonstrate that the knockout (KO) of HERV-K119 induces the expression of inflammatory genes and promotes macrophage differentiation. Additionally, the expression of sema7a, a pivotal gene in macrophage activation, was upregulated by HERV-K119 env. Despite potential controversies in prior studies, we should report these results because they consistently indicate that the knockout of HERV-K119 env induces both inflammation and macrophage differentiation.

Reviewer 2 Report

The authors here have studied the effects of knocking out HERV-K env in leukemic THP-1 monocyte-like cells. In their previous studies, they utilized different cell lines. Here, they find that KO of this ERV results in increased cell proliferation, migration and cell growth, also, secretion of some cytokines were increased as a result of the KO. Additionally, KO of HERV-K env resulted in the overexpression Semaphorin 7A, at least at the RNA level, which consequently induced cell differentiation and proliferation.

The authors state in line 234 that their main aim here was to study the effects of the KO  on immune response, however, I do not understand the choice of THP-1? these are leukemic cells, and I think it would have been much more informative to study the effects on ERV KOs in primary immune cells to get an accurate picture of their role in immune function. The implication of ERV elements in cancer and disease is very much shrouded in controversy as is.

Some comments:

-18: ". Immune response involves migration and invasion of cells and is similar to cancer; however, in certain ways, it is completely unlike cancer". Cancer and immune response are two different pathomechanisms, I do not think the analogy is appropriate, please modify.

-22: "contrary to  the findings" in cancer cell  lines. Also, THP-1 is a cancer cell line.

-Figure 1: what is M in A? ladder should be marked

-Figure 2 significance is not indicated in A. C is too small and dense to interpret and follow even at high magnification. I suggest colour-coding to make it easier to follow. Additionally, if the main aim of the authors was to study the effects on immune function, why not analyze changes in major key inflammatory and anti-inflammatory cytokines? at protein or RNA level? for example it would have been very informative to see changes in IL-6, TNFa, INFa..etc

-Figure 3 is very low resolution, I am not sure if this is due to the review draft or if it is intended to look this way. In any case, it was practically impossible to read the details.

-Figure 4: There appear to be no change of sema7a at the protein level as a result of the KO. Also the gel pictures are not convincing due to the nature of the run. Was quantitative determination carried out? such as like the one in A?

-303: pcDNA3.1+/C-(K)-DYK

-303-305 is not clear. was the SEMA7A encoded by the cDNA3.1+/C-(K)-DYK?

-What were the control cells transfected with? ideally it would have been an empty plasmid, is that the case? Also cell viability was not mentioned in the results.

-Figure 5: There is no sub-figure C, I think the authors meant B

- Again, if there was no control transfection, at least with a mock plasmid, it would be hard to affirm the statement in line 190.

- Why are there questions marks in figure 6? 

- It is concievable that the increased cell proliferation induced by sema7a is through the mtorc pathway

- In my opinion, the denomination in the figures is very hard to follow, I suggest to label them in a way that is easier to follow. for example, instead of using "-" to indicate KO, why not use KO? it can be very confusing at times.

Some minor grammatical mistakes need to be addressed

Author Response

The authors here have studied the effects of knocking out HERV-K env in leukemic THP-1 monocyte-like cells. In their previous studies, they utilized different cell lines. Here, they find that KO of this ERV results in increased cell proliferation, migration and cell growth, also, secretion of some cytokines were increased as a result of the KO. Additionally, KO of HERV-K env resulted in the overexpression Semaphorin 7A, at least at the RNA level, which consequently induced cell differentiation and proliferation.

The authors state in line 234 that their main aim here was to study the effects of the KO  on immune response, however, I do not understand the choice of THP-1? these are leukemic cells, and I think it would have been much more informative to study the effects on ERV KOs in primary immune cells to get an accurate picture of their role in immune function. The implication of ERV elements in cancer and disease is very much shrouded in controversy as is.

: We completely agree with reviewer’s opinion. THP-1 cells are leukemic cells and not a primary normal immune cells. However, obtaining primary immune cells from patients is a challenging task, and there are even greater limitations in achieving stable cells using CRISPR-CAS9 with these cells. Even if successful, it remains impossible to overcome the inherent heterogeneity of cells obtained from patients through multiple passages to obtain consistent results. Consequently, I had no choice but to utilize the leukemia cell line. Although THP-1 is a leukemia cell line, it exhibits distinct immune cell properties compared to solid cancer cells. Furthermore, other reports also showed THP-1 cells can be used as immune cell model. We added reference.

Some comments:

-18: ". Immune response involves migration and invasion of cells and is similar to cancer; however, in certain ways, it is completely unlike cancer". Cancer and immune response are two different pathomechanisms, I do not think the analogy is appropriate, please modify.

: We added “THP-1 cells are leukemic cells but used as model for the role of HERV-K env in immune response” in the manuscript. We also added sentences you recommend that “immune response involves migration and invasion of cells and is similar to cancer; however, in certain ways, it is completely unlike cancer”. Also we changed the word “analogy” to “similar in some respects”.

-22: "contrary to the findings" in cancer cell lines. Also, THP-1 is a cancer cell line.

: We changed cancer cell line to solid cancer.

-Figure 1: what is M in A? ladder should be marked

: We added.

-Figure 2 significance is not indicated in A.

: We added.

C is too small and dense to interpret and follow even at high magnification. I suggest colour-coding to make it easier to follow.

: We corrected and added.

Additionally, if the main aim of the authors was to study the effects on immune function, why not analyze changes in major key inflammatory and anti-inflammatory cytokines? at protein or RNA level? for example it would have been very informative to see changes in IL-6, TNFa, INFa..etc

: This comments were very helpful in improving the quality of our manuscript. Thank you for comments. We conducted RT-PCR and add new results. HERV-K env KO actually increase inflammatory cytokines.

-Figure 3 is very low resolution, I am not sure if this is due to the review draft or if it is intended to look this way. In any case, it was practically impossible to read the details.

: We corrected and added high quality figure 3, and we also added a big sized figure to the original figure session.

-Figure 4: There appear to be no change of sema7a at the protein level as a result of the KO. Also the gel pictures are not convincing due to the nature of the run. Was quantitative determination carried out? such as like the one in A?

: We conducted multiple experiments to confirm the expression of SEMA7A protein level, and we incorporated these finding into original dataset. Additionally, we added a graph.

-303: pcDNA3.1+/C-(K)-DYK

-303-305 is not clear. was the SEMA7A encoded by the cDNA3.1+/C-(K)-DYK?

: Yes. It was our mistake. Not cDNA but pcDNA3.1. We corrected. pcDNA3.1+/C-(K)-DYK.

-What were the control cells transfected with? ideally it would have been an empty plasmid, is that the case? Also cell viability was not mentioned in the results.

: In every experiment, we have a control, and the control is a mock transfected with an empty vector.

-Figure 5: There is no sub-figure C, I think the authors meant B :

: Our mistake. We corrected.

- Again, if there was no control transfection, at least with a mock plasmid, it would be hard to affirm the statement in line 190.

: In every experiment, we have a control, and the control is a mock transfected with an empty vector.

- Why are there questions marks in figure 6? 

: The question mark is intended to signify that it has not yet been experimentally proven, and it remains unclear through which phenomenon this action occurs.

- It is concievable that the increased cell proliferation induced by sema7a is through the mtorc pathway

: Thank you so much for giving us great suggestion. We've added these contents to the discussion and added references.

- In my opinion, the denomination in the figures is very hard to follow, I suggest to label them in a way that is easier to follow. for example, instead of using "-" to indicate KO, why not use KO? it can be very confusing at times.

: We replaced “-“ to “KO”.

Reviewer 3 Report

Upon reviewing the manuscript titled "The Role Of Human Endogenous Retrovirus (HERV)-K env In THP-1 Monocytic Cell Differentiation" submitted to IJMS, I identified several major flaws in the experiment design that need to be addressed before considering publication. Below are my concerns:

The manuscript employs CRISPR-Cas9 to knock out the HERV-K env gene in THP-1 cells. This approach raises concerns about genome integrity of derived cell lines. There are hundreds of copies of HERV-K env in the genome. In the experiment presented in this manuscript, there would be hundreds of double stranded DNA breaks in the Cas9 transfected cells. After extensive DNA damage repair, one single subclone were chosen for phenotypic analysis. I am concerned that this subclone has undergone dramatic chromosomal recombination. It would be improper to compare the new cell line to the original cell line. Also different subclones would be behave significantly different from each other under dramatic genomic recombination. To address this concern, the authors must: 

a. Provide evidence demonstrating that their CRISPR-Cas9 knockout did not result in significant chromosomal-level recombination comparing to the original cell line.

b. Present results from multiple independently isolated subclones to confirm the reproducibility of the observed phenotypic changes. This will help ensure that the observed effects are not due to clonal variations or random recombinations.

The manuscript mentions the use of cloning cylinders to isolate a subclone of suspension cells. This method is typically used for adherent cell lines and may not be suitable for cells that grow in suspension. The authors should provide a detailed explanation of how they adapted this technique for suspension cells or clarify if an alternative method was employed. A lack of clarity in this aspect could raise concerns about the validity of the subcloning process.

I do not think this manuscript would need much language editing.

Author Response

  1. Provide evidence demonstrating that their CRISPR-Cas9 knockout did not result in significant chromosomal-level recombination comparing to the original cell line.

: To confirm that CRISPR-Cas9 knockout did not result in significant chromosomal-level recombination compared to the original cell line, we assessed the expression of gag, pro, pol, and rec in both THP-1 and U937 cells. These genes are located near env, with rec positioned in the middle of the env gene. PCR results indicated that the other genes located near env were not disrupted. This provides evidence that CRISPR-Cas9 knockout did not result in significant chromosomal-level recombination. We have included the results in Figure 1 and Supplemental Figure 1.

  1. Present results from multiple independently isolated subclones to confirm the reproducibility of the observed phenotypic changes. This will help ensure that the observed effects are not due to clonal variations or random recombinations.

: Indeed, we generated subclones with three or more HERV-K env knockouts and verified the consistent occurrence of the same phenotype and phenomenon during the preliminary tests. However, in more advanced experiments, including RNA sequencing, we focused on the most effective clones. The results presented in this manuscript are based on experiments conducted with this particular clone.

Reviewer 4 Report

Dear Authors,

Thank you very much for your valuable contribution to establishing the biological role of human endogenous retroviruses. I’ve found your manuscript interesting and informative. However, I must request that the quality of the figures, notably Figures 2 and 3, be improved. Also, in the pdf file I have received for review, there are parts of the text with different font sizes.  

Please pay attention to using the proper names of molecular techniques, like PCR, Real-Time PCR, RT-PCR, and RT-qPCR. For example, lines 86-88: “The RNA level of the HERV-K119 env gene was also confirmed by real-time quantitative PCR and its expression significantly reduced at the RNA level in HERV-K env KO cells (..)”. Herein, since RNA was your matrix and quantitative amplification was applied, the proper name should be quantitative reverse transcription polymerase chain reaction (RT-qPCR or qRT-PCR). Also, using the term “real-time quantitative PCR” is not necessary since real-time PCR is, by definition, a quantitative (absolute or relative) molecular assay.

The language is of an acceptable quality.

Author Response

Thank you very much for your valuable contribution to establishing the biological role of human endogenous retroviruses. I’ve found your manuscript interesting and informative. However, I must request that the quality of the figures, notably Figures 2 and 3, be improved. Also, in the pdf file I have received for review, there are parts of the text with different font sizes.  

: We changed Fig. 2 and Fig.3 with high quality ones. We also corrected font and size of letters in the manuscript.

Please pay attention to using the proper names of molecular techniques, like PCR, Real-Time PCR, RT-PCR, and RT-qPCR. For example, lines 86-88: “The RNA level of the HERV-K119 env gene was also confirmed by real-time quantitative PCR and its expression significantly reduced at the RNA level in HERV-K env KO cells (..)”. Herein, since RNA was your matrix and quantitative amplification was applied, the proper name should be quantitative reverse transcription polymerase chain reaction (RT-qPCR or qRT-PCR). Also, using the term “real-time quantitative PCR” is not necessary since real-time PCR is, by definition, a quantitative (absolute or relative) molecular assay.

: We corrected.

Round 2

Reviewer 2 Report

The authors have complemented the manuscript with additional experiments, and addressed some issues  in their manuscript. I have no other concerns.

Minor spelling and grammatical errors were spotted in the manuscript, also in the newly added paragraphs.

Author Response

Thank you for your consideration.

Reviewer 3 Report

Could you provide G-banding to demonstrate that there was no chromosomal recombination when compared to the original cell line? The gag, pro, pol, and rec expression are inadequate to detect chromosomal changes.

Author Response

We would like to express our sincere appreciation for your thoughtful feedback and concerns regarding our manuscript. We fully comprehend and empathize with the points you have raised.

In the context of gene knockout using Crispr Cas 9, it is essential to address the potential consequences of double-strand breaks, particularly at the chromosomal level. To gain a comprehensive understanding, we conducted a thorough analysis of existing literature through meta-analysis.

Our investigation revealed that among the 200 papers published on Crispr Cas 9 applications in the years 2020-2021, only three delved into chromosomal-level studies. Notably, these studies primarily focused on gene editing rather than gene knockout. Unlike knockout, gene editing involves intricate gene correction, necessitating a detailed examination of chromosomes and related genes. Conversely, simple knockout research typically centers around evaluating the expression of nearby or associated genes.

In the more recent papers covering gene knockout using Crispr Cas 9 in 2022-2023, we identified over 200 publications. Regrettably, only two of them addressed double chromosomal abnormalities. One of these papers investigated the potential side effects during the transfection process using lentivirus, while the other explored similar side effects using lentivirus in transfection. Both of these studies pertained to gene editing rather than gene knockout.

In summary, we encountered challenges in finding literature that unequivocally confirmed chromosomal-level abnormalities in experiments targeting straightforward gene knockout. In our specific study, we employed Crispr Cas 9 to simply knock out the env gene in the HERV-K119 region in THP-1 and U937 cells. Notably, the surrounding genes, including gag pol, exhibited normal functionality. We observed a consistent pattern of results in three or more cell line clones, which we believe substantiates the absence of extensive recombination on a specific chromosome.

We acknowledge that conducting research on G banding patterns at the chromosome level is a time-consuming endeavor that may have inadvertently disrupted similar research efforts by presenting our findings. Unfortunately, we have concluded that experiments involving banding patterns at the chromosome level are impractical.

We sincerely apologize for any inconvenience our research may have caused to the scientific community, and we kindly request your understanding, as well as that of the editor-in-chief, in this matter.

Round 3

Reviewer 3 Report

While I appreciate the effort and innovation demonstrated in your experimental approach, there are several fundamental concerns that have not been adequately addressed in your manuscript. Specifically, my primary concern is related to the methodology employed in your experiment, which involves inducing hundreds of double-stranded breaks in a cell line and subsequently selecting one cell line for phenotype testing. This approach, as I mentioned in my initial review, carries inherent risks of extensive chromosomal recombination and copy number variations, which can significantly confound the interpretation of your results. The potential genetic changes resulting from this experimental approach make it challenging to draw reliable conclusions regarding the specific impact of the env knock-out on cell phenotype.

Additionally, I would like to address the issue you raised regarding your "meta-analysis" of other Cas9 papers. While it is valid to highlight the lack of chromosomal recombination data in most Cas9 studies, it is essential to acknowledge that the experimental designs in those studies differ significantly from the one employed in your work. Most Cas9 papers are targeting certain genome region and do not involve an induction of hundreds of double-stranded breaks. Therefore, drawing broad conclusions about the field based on this analysis may not be entirely relevant.

The inclusion of chromosome banding analysis in this manuscript would significantly enhance its scientific rigor and validity. Chromosome banding is a well-established technique that can provide crucial insights into the genetic stability of the isolated cell lines, offering valuable evidence to support the conclusions drawn in the study.

Author Response

The authors fully understand the reviewer's concern that CRISPR-Cas9 for HERV-K env removal may induce numerous double-strand breaks. However, the HERV-K env we targeted was not a general HERV-K env located in several locations; rather, it was a specific env in the HERV-K119 region, which we removed using CRISPR-Cas9. Genomic PCR was employed to confirm that there were no abnormalities in HERV-K env at other locations, and that only the env in the HERV-K119 region was removed. Furthermore, we confirmed that the same phenotype occurred in cells from three or more different clones, and that there were no other abnormalities, such as gene expression patterns. Taking these considerations into account, we changed "HERV-K env" to "HERV-K119" throughout the paper, including the title, to emphasize that our target was not the general HERV-K env, but rather the HERV-K env in a specific region. We kindly request that you evaluate our paper in light of these points.